# Treatment Refractoriness in Chronic Lymphocytic Leukemia: Old and New Molecular Biomarkers

**DOI:** 10.3390/ijms241210374

**Published:** 2023-06-20

**Authors:** Nawar Maher, Samir Mouhssine, Bassam Francis Matti, Alaa Fadhil Alwan, Gianluca Gaidano

**Affiliations:** 1Division of Hematology, Department of Translational Medicine, Università del Piemonte Orientale and Azienda Ospedaliero-Universitaria Maggiore della Carità, 28100 Novara, Italy; 20024416@studenti.uniupo.it (N.M.); 20023477@studenti.uniupo.it (S.M.); 2Department of Hematology and Bone Marrow Transplant, Hematology and Bone Marrow Transplant Center, Baghdad 00964, Iraq; bassam_francis@yahoo.com; 3Department of Clinical Hematology, The National Center of Hematology, Mustansiriyah University, Baghdad 10015, Iraq; ala_sh73@uomustansiriyah.edu.iq

**Keywords:** chronic lymphocytic leukemia, predictive biomarkers, chemoimmunotherapy, pathway inhibitors, immunotherapy

## Abstract

Chronic lymphocytic leukemia (CLL) is the most common leukemia in adults. Despite its indolent clinical course, therapy refractoriness and disease progression still represent an unmet clinical need. Before the advent of pathway inhibitors, chemoimmunotherapy (CIT) was the commonest option for CLL treatment and is still widely used in areas with limited access to pathway inhibitors. Several biomarkers of refractoriness to CIT have been highlighted, including the unmutated status of immunoglobulin heavy chain variable genes and genetic lesions of *TP53*, *BIRC3* and *NOTCH1*. In order to overcome resistance to CIT, targeted pathway inhibitors have become the standard of care for the treatment of CLL, with practice-changing results obtained through the inhibitors of Bruton tyrosine kinase (BTK) and BCL2. However, several acquired genetic lesions causing resistance to covalent and noncovalent BTK inhibitors have been reported, including point mutations of both *BTK* (e.g., C481S and L528W) and *PLCG2* (e.g., R665W). Multiple mechanisms are involved in resistance to the BCL2 inhibitor venetoclax, including point mutations that impair drug binding, the upregulation of BCL2-related anti-apoptotic family members, and microenvironmental alterations. Recently, immune checkpoint inhibitors and CAR-T cells have been tested for CLL treatment, obtaining conflicting results. Potential refractoriness biomarkers to immunotherapy were identified, including abnormal levels of circulating IL-10 and IL-6 and the reduced presence of CD27^+^CD45RO^−^ CD8^+^ T cells.

## 1. Introduction

Chronic lymphocytic leukemia (CLL) is a hematologic neoplasm characterized by the clonal proliferation of mature B cells [1,2]. Although leukemia was first described in 1845 by R. Virchow and J.H. Bennett, it was only in 1924 that CLL was fully characterized by G.B. Minot and R. Isaacs as a distinct clinical entity [3]. However, CLL has been extensively studied in more recent times, with the identification of key genetic abnormalities implied in its pathogenesis, such as deletion 11q, deletion 13q, and trisomy 12 [4]. For a clinical diagnosis, a lymphocyte blood count of ≥5 × 10^9^/L is required, as well as a specific immunophenotypic profile that necessarily includes the clonal expression of CD19, CD5, CD20, CD23 and dim surface immunoglobulins with κ or λ light-chain restriction [1,2,5]. CLL is the most common leukemia in adults, with an incidence of 4.6/100,000 per year in the US, and is characterized by a relatively low mortality, displaying a 5-year relative survival rate of nearly 90% [6]. The median age at diagnosis is ~70 years, implying that CLL is a disease mainly of older adults [6].

Despite its indolent clinical course, CLL is still not curable, and patients refractory to available therapies are prone to disease progression or transformation to aggressive lymphoma and eventually death, highlighting the need to overcome treatment refractoriness in this disease [7,8]. The staging of CLL is based on the historically used scores of the Binet and Rai systems that are based on clinical variables, namely anemia, thrombocytopenia, hepatosplenomegaly and lymphadenopathy [9,10]. In recent times and thanks to the availability of novel biomarkers, several prognostic scoring systems have been developed, such as the CLL International Prognostic Index (CLL-IPI) and the International Prognostic Score for Asymptomatic Early-Stage Disease (IPS-E), allowing a more precise prognostic assessment of patients on the basis of clinical and molecular features [11,12].

In asymptomatic patients with early-stage disease (Rai 0/Binet A), which account for the majority of CLL cases, a watch-and-wait approach should be adopted, while treatment requirement at diagnosis or during follow-up is based on specific indications according to the International Workshop on Chronic Lymphocytic Leukemia (iwCLL) guidelines [13,14]. Remarkably, 2–10% of CLL patients develop an aggressive lymphoma, defined as Richter syndrome (RS), which is particularly refractory to chemoimmunotherapy (CIT) and still represents a life-threatening condition and a major unmet clinical need [15]. 

Until the advent of pathway inhibitors, the front-line treatment for CLL was represented by CIT, in particular the FCR (fludarabine, cyclophosphamide and the anti-CD20 monoclonal antibody [mAb] rituximab), BR (bendamustine and rituximab) and Chl-O (chlorambucil and the anti-CD20 mAb obinutuzumab) regimens [13]. CIT has now been largely abandoned due to the lower efficacy and higher incidence of adverse events compared to pathway inhibitors [16,17]. CIT can still be evaluated as a treatment option in chemo-sensitive patients, especially in less-resourced countries, where pathway inhibitors are hardly available or affordable [18,19,20,21]. 

Current front-line treatment for CLL is based on pathway inhibitors with or without anti-CD20 mAbs. The European Society for Medical Oncology (ESMO) guidelines for front-line treatment of CLL include Bruton tyrosine kinase inhibitors (BTKi), namely ibrutinib and acalabrutinib, the B-cell lymphoma 2 inhibitor (BCL2i) venetoclax with or without obinutuzumab, and the phosphoinositide 3 kinase inhibitor (PI3Ki) idelalisib with rituximab, although the latter regimen is seldom used because of infectious toxicity [18]. Consistently, the latest National Comprehensive Cancer Network (NCCN) guidelines recommend the use of a BTKi in monotherapy (ibrutinib or zanubrutinib) or a combination of acalabrutinib or venetoclax plus obinutuzumab for the front-line treatment of CLL [21]. For relapsed/refractory (R/R) CLL, the guidelines recommend the use of one of the above-mentioned pathway inhibitors based on response to front-line therapy; several innovative drugs are also under development [18,21]. Although allogeneic hematopoietic stem cell transplantation (allo-HSCT) is the only curative option for CLL, with a 5-year progression-free survival (PFS) rate of ~40%, it is rarely used since the procedure is affected by a significant mortality rate (approximately from 10% to 20%), primarily due to graft-versus-host disease (GVHD) and its complications [22,23]. As different treatment options were progressively utilized for the treatment of CLL, several biomarkers of refractoriness have been identified (Table 1). Here, we aim to provide a comprehensive overview of the biomarkers of CLL refractoriness to available therapies, including CIT as well as pathway inhibitors and immunotherapy.

## 2. Biomarkers of Refractoriness to Chemoimmunotherapy

Historically, chemotherapy had been the most widely used option for the treatment of CLL, which was subsequently replaced by CIT based on the results of practice-changing clinical studies [49]. The most used chemotherapy regimens were initially based on monotherapy with alkylating agents, such as chlorambucil and bendamustine, or purine analogues, namely fludarabine and cladribine [4]. As for combination therapies, the most adopted was the FC regimen (fludarabine and cyclophosphamide), which granted favorable results compared to monotherapies [4]. CIT for CLL consists of the combination of chemotherapy and anti-CD20 mAbs in order to obtain a synergistic effect against tumor cells [13]. The CLL8 phase III randomized trial compared the chemotherapy regimen FC versus FCR in fit CLL patients, demonstrating a significant superiority of the chemoimmunotherapeutic approach [27,50]. Specifically, FCR outperformed FC in both median PFS (56.8 vs. 32.9 months, respectively) and overall survival (OS, not reached vs. 86.0 months, respectively) with a comparable toxicity profile [27].

Different CIT regimens for CLL patients with comorbidities were evaluated in the CLL11 phase III randomized trial, where the Chl-O regimen showed better median PFS and OS compared to chlorambucil plus rituximab or chlorambucil monotherapy [28,51]. The adoption of CIT regimens for CLL has led to the identification of several biomarkers of refractoriness to this therapeutic approach, including the unmutated status of immunoglobulin heavy-chain variable (IGHV) genes and genetic lesions of *TP53*, *BIRC3* and *NOTCH1* [24]. In order to overcome resistance to CIT, pathway inhibitors were adopted for the treatment of CLL, with practice-changing results obtained by BTKi and BCL2i [49].

### 2.1. IGHV Mutational Status

Mature B cells express the B cell receptor (BCR) on their surface, a key component for antigen recognition and B cell activation, composed of an immunoglobulin (Ig) and a signaling subunit [52]. In order to expose an Ig on the external cell membrane, B cells must perform a genetic recombination of the variable Ig genes through a process termed V(D)J rearrangement, which ensures a very high degree of heterogeneity in the BCR repertoire [53]. After antigen recognition, naïve B cells move to lymph node germinal centers (GCs), where somatic hypermutation (SHM) of IGHV genes takes place, potentially increasing the BCR affinity for the recognized antigen [53,54].

Based on the mutational status of IGHV genes, CLL can be divided into two molecular subgroups: (i) IGHV unmutated CLL (U-CLL, ~40% of all CLL), which reflect mature B cells that have not experienced the GC reaction and have undergone maturation in a T-cell-independent manner; and (ii) IGHV mutated CLL (M-CLL, ~60% of all CLL), which reflect mature B cells that have experienced the GC reaction and have undergone the SHM process [25,55,56]. In particular, to be considered as M-CLL, the threshold used in the clinical practice is a deviation in ≥2% of the patient’s IGHV sequence from the germline nucleotide sequence [57]. Unmutated IGHV genes associate more commonly with progressive or R/R CLL, while mutated IGHV genes are more frequently detected in asymptomatic or treatment-naïve disease [58,59]. Importantly, unmutated IGHV genes occur in up to 60% of CLL, requiring treatment according to guidelines. 

Beyond its prognostic value, IGHV mutational status is also a predictive biomarker, as shown by the lower response of U-CLL to all the available CIT regimens when compared to M-CLL [27,60,61,62]. Clinical trials evaluating continuous treatment with ibrutinib, acalabrutinib and zanubrutinib have displayed favorable efficacy outcomes in U-CLL, superimposable to those reached with M-CLL, overcoming treatment refractoriness due to IGHV mutational status [63,64,65].

### 2.2. TP53 Disruption

*TP53*, located on the short arm of chromosome 17 (17p), is an onco-suppressor gene encoding the p53 protein, also called “the guardian of the genome”, which exerts a proapoptotic function in response to DNA damage [66]. Consistently, the disruption of *TP53* results in increased resistance to apoptosis induced by DNA-damaging agents, including chemotherapy and, by extension, CIT [67]. Somatic mutations are the most common genetic lesions of *TP53* in CLL, followed by del(17p) [68,69].

The disruption of *TP53* has been found in 4% to 8% of newly diagnosed CLL, while, as the disease progresses, the frequency of *TP53* abnormalities rises, reaching a prevalence of 10–12% at the time of first treatment requirement, ~40% in patients refractory to fludarabine, and 50–60% in those who develop RS [69]. Consequently, genetic lesions of *TP53* can be defined as both prognostic and predictive biomarkers. Among patients treated with FCR, the CLL8 trial reported a median PFS of 15.4 months and a median OS of 49 months in *TP53*-mutated patients, while the median PFS and OS in *TP53* wild-type patients were 59 months and not reached, respectively [27]. Similar unfavorable outcomes were reached with the BR and Chl-O regimens in *TP53*-disrupted CLL, while BTKi-based therapies have obtained remarkable results, which are comparable with those of *TP53* wild-type patients [27,28,61,63,64,65].

Due to the significant clinical impact exerted by *TP53* disruption, the iwCLL guidelines recommend testing del(17p) via fluorescence in situ hybridization (FISH) and *TP53* mutation status via DNA sequencing before every line of treatment [14]. In addition to these recommendations, the European Research Initiative on CLL (ERIC) endorses the possible use of next-generation sequencing (NGS) for *TP53* mutation testing since this methodology is characterized by a higher sensitivity compared to traditional Sanger sequencing [70].

### 2.3. BIRC3 Disruption

The *BIRC3* gene has been found to be mutated or deleted in 2–6% of CLL cases [31,32,33]. *BIRC3* encodes for the protein c-IAP2, which negatively regulates the MAP3K14 kinase (or NIK–NF-κB-inducing kinase), the key activator of the noncanonical NF-κB signaling pathway, leading to the transcription of genes linked to cell proliferation and survival [71]. Additionally, the aberrant activation of NF-κB signaling in c-IAP2 knockdown models has been shown to increase p53 degradation via the E3 ubiquitin ligase MDM2 [72]. Therefore, the mutational inactivation or deletion of *BIRC3* in CLL results in constitutive NF-κB pathway activation, providing pro-survival signals to the leukemic clone, e.g., through the up-regulation of several anti-apoptotic genes, as demonstrated in ex vivo models [34,73].

A retrospective evaluation of the outcome of FCR-treated CLL patients showed a similar median PFS rate between *BIRC3*- and *TP53*-disrupted CLL patients (2.2 and 2.6 years, respectively), significantly inferior to the PFS of *BIRC3* wild-type patients [34]. Moreover, the CLL14 phase III randomized trial, evaluating front-line treatment with venetoclax plus obinutuzumab versus Chl-O in CLL, has shown poor outcomes in *BIRC3* mutated patients treated with Chl-O, with a median PFS of 16.8 months [35,74]. However, ibrutinib- and/or venetoclax-based therapies appear to overcome the resistance conferred by *BIRC3* disruption [34,74,75,76,77].

### 2.4. NOTCH1 Mutations

*NOTCH1* codes for the transmembrane protein NOTCH1, which acts as a surface receptor for ligands of the SERRATE/JAGGED or DELTA families [78,79]. After being cleaved by γ-secretase, the active subunit of the receptor migrates into the nucleus and acts as a transcription factor for genes involved in cell survival and proliferation, including *MYC* and components of the NF-κB pathway [37,78]. In CLL, *NOTCH1* mutations disrupt the PEST domain, responsible for the promotion of proteasomal degradation of the NOTCH1 protein, resulting in the aberrant activation of the receptor [80,81].

At diagnosis, ~8% of CLL patients harbor a *NOTCH1* mutation, but the prevalence of this genetic lesion rises in fludarabine-refractory CLL and RS patients (20.8% and 31.1%, respectively) [38]. Furthermore, genetic lesions of *NOTCH1* are thought to be involved in resistance to CLL immunotherapy [79]. The predictive value of *NOTCH1* mutations for the treatment with an immunotherapeutic agent has been investigated in the above-mentioned CLL8 trial [27]. In particular, *NOTCH1*-mutated CLL showed no improvement due to the addition of rituximab since the 5-year PFS rate of *NOTCH1*-mutated patients was 25.8% for the FC cohort and 26.7% for the FCR cohort (*p*-value of 0.974) [27]. Similar results were obtained by the COMPLEMENT 1 study, a phase III randomized trial that compared chlorambucil alone versus chlorambucil plus the anti-CD20 mAb ofatumumab, highlighting the role of *NOTCH1* mutations in predicting refractoriness to anti-CD20 mAb-based immunotherapy [39,82].

The proposed mechanism for *NOTCH1*-mediated resistance appears to be linked to the HDAC-mediated repression of the surface exposure of CD20 in *NOTCH1*-mutated CLL cells, as shown by in vitro models [83]. Remarkably, the CLL11 trial demonstrated a clear superiority of Chl-O over chlorambucil alone, showing better PFS and OS in the Chl-O arm, independent of the presence of *NOTCH1* genetic lesions [28]. Hence, these data suggest that the higher clinical efficacy of obinutuzumab may overcome the effect of *NOTCH1* mutations. Pathway inhibitors represent a viable option to overcome NOTCH1-mediated refractoriness to CIT and immunotherapy, as shown in the RESONATE phase III randomized trial, where no difference in PFS was detected between *NOTCH1* mutated and wild-type CLL treated with ibrutinib [63].

## 3. Biomarkers of Refractoriness to BTK Inhibitors

The BCR signaling pathway plays an essential role in the development of B cells and the pathogenesis of CLL [84]. BCR signaling is activated by antigen binding to surface immunoglobulins (sIg), resulting in coupling and autophosphorylation of the immunoreceptor tyrosine-based activation motifs (ITAMs) on the cytoplasmic tails of CD79A (Igα)/CD79B (Igβ) by the Src family protein kinase LYN. The phosphorylation of ITAMs creates docking sites for spleen tyrosine kinase SYK, which activates the B cell linker scaffold protein (BLNK). In the presence of BLNK, BTK is activated through phosphorylation at its Y551 aminoacidic residue by either LYN or SYK [85]. Once activated, BTK triggers downstream pathways, namely the PI3K-Akt and phospholipase-γ-2 (PLCγ2) pathways, finally leading to the induction of different transcription factors, including mTOR, NF-κB, ERK1/2, and NFAT, and it is involved in the survival, differentiation, and proliferation of B cells [85,86]. Additionally, BTK can be triggered by other receptors, including growth factors, cytokine receptors, and G-protein coupled receptors (GPCRs), such as chemokine receptors and integrins [87]. In CLL, the BCR is constitutively active through ligand-dependent and independent mechanisms, causing constitutive BTK signaling activation, which confers a survival and proliferation advantage to the neoplastic cells (Figure 1) [88].

### 3.1. Refractoriness to First-Generation Covalent BTK Inhibitors

Ibrutinib, the first-in-class and orally bioavailable BTKi, was approved by the FDA in 2014 for the treatment of R/R CLL, changing the landscape for the treatment of this leukemia. Ibrutinib can be administrated both as a first-line therapy and in R/R CLL patients [86]. Its safety and efficacy as front-line treatment were established by the randomized phase III RESONATE-2 trial, which documented the superiority of ibrutinib versus chlorambucil [89]. In R/R CLL, ibrutinib was proven to be superior to the anti-CD20 monoclonal antibody ofatumumab by the randomized phase III RESONATE trial [90].

Ibrutinib irreversibly blocks BTK by covalently binding to the C481 aminoacidic residue in the ATP-binding domain of the protein [91]. The occupancy of the ATP-binding site by ibrutinib leads to a lack of phosphorylation of different downstream targets such as Akt and PLCγ2, resulting in BTK signaling inhibition that, in turn, reduces BCR signaling both in vitro and in vivo [92]. Besides this on-target effect, ibrutinib deactivates several off-targets, e.g., EGFR, ErbB2, ITK, and TEC, which might contribute to the antitumor effect but, at the same time, result in adverse events, such as atrial fibrillation (AF) and bleeding [93,94].

#### 3.1.1. Primary Resistance to First-Generation Covalent BTK Inhibitors

Biologic resistance can be divided into primary (or intrinsic) and secondary (or acquired) resistance [95]. Primary resistance to ibrutinib can develop in 10–16% of the cases, but the molecular mechanisms are still unclear [96,97]. Resistance to ibrutinib in CLL patients has been associated with high-risk genomic features and heavy pretreatment. Genetic alterations unrelated to the BCR pathway, involving the ATM, BIRC3, NOTCH1, SF3B1, and TP53 genes, are found in a fraction of treatment-naïve CLL patients [96,97].

Importantly, baseline features, such as del(17p)/*TP53* and complex karyotype (≥3 chromosomal abnormalities), increase the risk of disease progression in patients treated with ibrutinib [29,98,99]. *TP53* disruption is the only independent molecular factor that predicts inferior OS and PFS in patients receiving ibrutinib treatment [100]. Therefore, despite major improvements compared to the CIT era, the prognosis of patients with del(17p)/*TP53* remains suboptimal, at least to a certain extent, even in the era of targeted therapy. Additionally, del(18p), which occurs in a small percentage of untreated cases, is found at a higher frequency in ibrutinib-relapsed patients and is associated with BTK mutation development, suggesting a potential role of this chromosomal abnormality in refractoriness to ibrutinib [29].

#### 3.1.2. Secondary Resistance to First-Generation Covalent BTK Inhibitors

Secondary resistance eventually develops in about 60% of CLL patients treated with BTK covalent inhibitors [42]. Shortly after ibrutinib was approved in 2014, the first BTKi resistance mutations were reported. These mutations can be divided into two categories: (i) point mutations that prevent ibrutinib from binding covalently to BTK by changing the targeted cysteine residue (C481) in the kinase domain; and (ii) mutations that constitutively activate downstream signaling through PLCγ2 (Figure 1) [101]. 

In an early analysis, Woyach and colleagues performed whole-exome sequencing in six patients with acquired resistance to ibrutinib and documented that four of them had a cysteine-to-serine mutation in *BTK* at position 481 (C481S), corresponding to the ibrutinib binding site [43]. This pivotal investigation also revealed that ibrutinib refractoriness exploited at least another molecular mechanism. In fact, one patient with a low-frequency C481S mutation in *BTK* harbored three different *PLCG2* mutations: R665W, L845F, and S707Y. The sixth patient had an arginine-to-tryptophan mutation in *PLCG2* at position 665 (R665W). In all patients, the mutations identified at the time of refractoriness were absent before the start of treatment [43]. 

The C481S mutation was later confirmed to be the most common ibrutinib resistance-mediating mutation in CLL patients [102]. The functional characterization of the C481S mutation has shown reduced affinity of BTK for ibrutinib, allowing only reversible inhibition rather than irreversible blockade [43,103]. In addition to the substitution of cysteine in 481 by serine, the substitution by other amino acids such as tyrosine, arginine, phenylalanine, tryptophan, and glycine has also been reported [42,99]. Besides mutations of BTK, also the *PLCG2* gene, which acts downstream of BTK in the BCR signaling cascade, is involved in resistance to ibrutinib. In particular, the R665W mutation in the SH2 domain of PLC*γ*2 is a gain-of-function mutation that causes activation of PLC*γ*2 independent of *BTK* signaling stimulation [104]. 

Deep sensitivity genetic analysis of sequential samples of progressive CLL patients on ibrutinib allowed the detection of *BTK*- and/or *PLCG2*-mutated clones at a median of eight and nine months before progression, respectively [105,106]. More recently, an analysis of a French registry cohort showed an incidence of *BTK* mutations of 57% in patients who were still responding to ibrutinib, suggesting that a substantial proportion of CLL patients receiving ibrutinib monotherapy already have resistance-mediating mutations though still responding clinically to treatment [42]. In this respect, *BTK* and/or *PLCG2* mutations might represent potential biomarkers to detect preclinical resistance development to BTKi, although current guidelines do not recommend testing for *BTK* and *PLCG2* mutations before the development of clinical refractoriness. Remarkably, only approximately 70–80% of patients with acquired ibrutinib resistance carry mutations in *BTK* and/or *PLCG2* [40]. Several chromosomal aberrations have also been associated with secondary resistance to ibrutinib, including *MYC* amplification, del(18p), and del(8p), which causes haploinsufficiency of *TRAIL-R*, generating resistance of the neoplastic cells to TRAIL-induced apoptosis [29,34,99,107].

### 3.2. Refractoriness to Second-Generation Covalent BTK Inhibitors

Acalabrutinib and zanubrutinib are irreversible, potent and covalent BTK inhibitors with higher selectivity than ibrutinib for the C481 residue of the binding site [108,109]. Therefore, these drugs have less off-target inhibition of other kinases of the TEC family, including EGFR and ITK, and consequently, fewer adverse events [108,109]. The phase III randomized ELEVATE-RR trial showed the non-inferiority of acalabrutinib compared to ibrutinib in terms of PFS in R/R CLL [108]. Recently, a phase III randomized controlled trial comparing ibrutinib and zanubrutinib in R/R CLL patients showed that zanubrutinib had a higher ORR, superior PFS, and a lower rate of atrial fibrillation/flutter compared with ibrutinib [109]. However, secondary resistance to second-generation BTKi has been reported as well. In patients treated with acalabrutinib, the C481S mutation was found to be the most common acquired mutation in disease progression, similar to ibrutinib [110]. Moreover, the development of *PLCG2* mutations was also detected in the same cohort [110]. Conversely, in CLL patients with progressive disease treated with zanubrutinib, the BTK L528W mutation is detected at a certain rate and appears to be responsible for progression (Figure 1) [111].

### 3.3. Strategies to Overcome Resistance to Covalent BTK Inhibitors

Point mutations of C481 in the ATP-binding pocket of BTK are responsible for a sizable fraction of CLL resistance to ibrutinib [40]. To overcome this resistance, reversible and non-covalent BTK inhibitors have been developed, including vecabrutinib, fenebrutinib, nemtabrutinib (ARQ 531), and pirtobrutinib (LOXO-305) [112,113,114]; these compounds have demonstrated to be effective on both C481-mutated and unmutated BTK in preclinical studies. Remarkably, the BRUIN phase I/II trial has shown that pirtobrutinib achieves an ORR of 62% in CLL R/R to multiple lines of treatment, the majority of which had been previously treated with a covalent BTKi [115]. On these grounds, non-covalent BTKi offer a new frontier for CLL patients that are refractory to other pathway inhibitors, including covalent BTKi. Despite these clinically important advances, several mutations causing acquired resistance to non-covalent BTKi and some covalent BTKi have been recently reported, including point mutations in the tyrosine kinase domain of BTK such as V416L, A428D, M437R, T474I, and L528W [41]. Functional analysis of these mutations indicates that these amino acid changes impair BTK binding to both non-covalent and covalent BTKi [41]. Consistently, Blombery et al. recently described an enrichment of the BTK-L528W mutation in patients receiving the covalent BTKi zanubrutinib compared to patients treated with ibrutinib, raising the possibility of cross-resistance between zanubrutinib and the novel reversible and non-covalent BTKi (Figure 1) [116].

Different strategies to overcome the above-cited resistance to BTKi are currently under investigation. In particular, concerning ibrutinib-resistant CLL associated with *PLCG2* mutations, in vitro studies have shown that the inhibition of SYK and LYN, both necessary for the activation of PLCγ2 independently of BTK, can overcome sustained survival signaling [104]. In a recent phase II study, the SYK inhibitor entospletinib resulted in a response in patients previously treated with inhibitors of the B cell receptor pathway, even in patients who had BTK and *PLCG2* mutations. However, the ORR was low (33%), and PFS was 5.56 months [117].

Furthermore, proteolysis-targeting chimeras (PROTACs) may be a new approach to overcome BTKi resistance. PROTACs were shown to be effective against in vitro mutant BTK-C481 cells by inducing BTK degradation through ubiquitin-mediated protein degradation [118]. NX-2127 is the first-in-class targeted protein degrader of BTK, which in a preclinical study, was shown to induce the degradation of both wild-type and mutant BTK [119]. The clinical results of a first-in-human phase I trial on NX-2127 were reported recently. A total of 23 R/R CLL patients with a median of six prior therapies (2–11) were enrolled in the study. All patients had been previously treated with a covalent BTKi and/or venetoclax. This patient group, for whom no other therapeutic options were available, NX-2127, resulted in an ORR of 33% in 12 evaluable patients at a median follow-up of 5.6 months. These data support the use of BTK degraders in double- or triple-refractory patients, regardless of BTK or BCL2 mutation status [119].

## 4. Biomarkers of Refractoriness to BCL2 Inhibitors

Mitochondrial apoptosis is controlled by the BCL2 family, which includes pro-apoptotic proteins (Bak and Bax), anti-apoptotic proteins (namely BCL2, BCL-xL, and MCL1) and BH3-only proteins (namely BIM, BID, BAD, PUMA, BIK, and NOXA) [120]. In normal cells, anti-apoptotic proteins play their role by binding and sequestering the mitochondria pore-forming proteins Bak and Bax. Upon cellular stress, BH3-only proteins bind to anti-apoptotic proteins via their BH3 domain, releasing pore-forming proteins. These pore-forming proteins bind to the outer mitochondrial membrane to trigger membrane permeability and the release of cytochrome C from mitochondria, allowing the assembly and activation of the apoptosome, which leads to the activation of a caspase cascade that initiates apoptosis (Figure 2) [120].

Different aberrant mechanisms lead to the overexpression of BCL2 in the primary phase of CLL tumorigenesis, among which the loss of miR-15 and miR-16 at 13q14, which is detectable in 40–60% of CLL patients and hypomethylation of the BCL2 gene [121,122]. The miR-15 and miR-16 miRNAs physiologically inhibit the translation of the BCL2 protein by binding to a specific sequence on the corresponding mRNA. In CLL, the loss of these two miRNAs leads to increased levels of BCL2, providing a survival advantage for the tumor [121,122].

Venetoclax (formerly known as ABT-199) is a first-in-class, orally administrable, BH-3 mimetic drug designed to have high affinity and selectivity for BCL2 and a low affinity for MCL1 and BCL-xL, which are crucial for platelet survival [123]. Venetoclax was initially approved in 2016 as a monotherapy for relapsed CLL patients with del(17p) or *TP53* mutation or who are not suitable for BCR inhibitors, as well as for patients without del(17p) or *TP53* mutation and refractory to CIT and BCR pathway inhibitors. Despite the major clinical achievements obtained with the combination of venetoclax and an anti-CD20 mAb, a fraction of patients fail therapy and progress [35,124]. In that regard, long-term venetoclax treatment can eventually result in the expansion of resistant clones and progression, driven by clonal evolution [125]. Multiple mechanisms have been proposed to explain venetoclax resistance, including reduced drug binding due to gene mutations, upregulation of BCL2-related anti-apoptotic family members, and microenvironmental alterations [120].

### 4.1. Primary Resistance to BCL2 Inhibitors

Primary therapeutic resistance to venetoclax has been associated with intra-tumoral heterogeneity and clonal evolution. Pre-existent mutations in CLL sub-clones likely contribute to resistance by conferring a certain growth advantage or access to supportive microenvironment niches [126,127]. Primary venetoclax resistance may also be due to epigenetic mechanisms, including DNA methylation, post-translational histone modifications, and chromatin remodeling [128]. These alterations regulate the growth rate and response to environmental pressures, which ultimately influence tumor heterogeneity and clonal evolution [128]. On top of the mechanisms mentioned above, various microenvironmental signals (IL-10, CD40L, etc.) also contribute to intrinsic resistance by stimulating TLR9, that in turn activates NF-κB signaling [45]. Importantly, the activation of the transcription factor NF-κB leads to increased expression of the anti-apoptotic proteins BCL-xL and MCL1 [45].

### 4.2. Secondary Resistance to BCL2 Inhibitors

In a similar way to ibrutinib resistance, point mutations causing venetoclax resistance have also been reported. The first identified and most common mutation is G101V in the BH3-binding groove of BCL2, leading to refractoriness in about 15% of CLL patients treated with venetoclax [44]. This mutation reduces by 180-fold the affinity of BCL2 for venetoclax, preventing the drug from displacing proapoptotic BH-3-only proteins (e.g., BIM) from BCL2 [129]. In addition, the D103Y and F104I mutations were also found to cause drug resistance (Figure 2) [130]. Interestingly, both G101V and D103Y were detectable in several patients before the occurrence of clinical relapse, which raises the hypothesis that, in the future, these genetic changes might be considered as predictive biomarkers of treatment failure and lead to early intervention, e.g., by the addition of other therapeutic agents, such as BTKi [131]. 

The fact that BCL2 mutations were found only in a subset of patients suggested the involvement of other mechanisms leading to venetoclax resistance [129,130]. It is well established that the overexpression of the anti-apoptotic BCL-xL and MCL1 is associated with a higher risk of drug resistance [46]. Consistently, Ghia and colleagues found that higher expression of ROR1 before and after one year of venetoclax treatment is associated with accelerated disease progression and shorter OS regardless of IGHV mutation status. Notably, the increased expression of ROR1 is accompanied by the upregulation of WNT5a-ROR1 signaling, leading to the higher expression of ERK1/2 and NF-κB-target genes, including the BCL-xL protein, which may enhance venetoclax resistance since it inhibits apoptosis and is not significantly targeted by venetoclax [132]. Furthermore, the amplification of 1q23 encompassing *MCL1* and *PRKAB2* (a component of the AMPK pathway) and the overexpression of these two genes have been demonstrated in patients with venetoclax resistance [46]. 

Curiously, whole-exome sequencing and methylation profiling in a cohort of CLL patients before venetoclax treatment and at the time of venetoclax resistance revealed no genetic alterations in *BCL2* [30]. However, most patients developed mutations in other cancer-related genes, including *BRAF*, *NOTCH1*, *RB1*, and *TP53* or had a homozygous deletion of *CDKN2A/B*, suggesting a potential mechanism of resistance that involves the deregulation of these genes [30]. Lastly, the overexpression of NOTCH2 has also been potentially implicated as a novel mechanism of venetoclax resistance. In fact, Fiorcari et al. have recently reported that CLL patients harboring trisomy 12 have high levels of NOTCH2, leading to the upregulation of MCl-1, which in turn promotes cell survival by evading the proapoptotic effect of venetoclax [133].

### 4.3. Strategies to Overcome Resistance to BCL2 Inhibitors

Because BTKi and BCL2i inhibit different biological pathways, an obvious clinical strategy is to treat venetoclax-resistant patients with a BTKi. In two retrospective series with similar results, BTKi achieved an ORR of 84% to 91% and a median PFS of 32 to 34 months for patients with progressive disease (PD) after venetoclax, including those harboring *BCL2* mutations [134,135]. Patients with prior BTKi intolerance who had developed venetoclax resistance could achieve durable remissions with BTKi rechallenging, preferentially with an alternative agent compared to the BTKi used previously in their clinical course [135]. In contrast, PI3Ki are associated with poor outcomes after venetoclax (median PFS, 5 months) [136].

Recent studies have shown that targeting epigenetic mechanisms with HDAC inhibitors, DNA methyltransferase inhibitors, or bromodomain reader protein inhibitors can downregulate the gene expression signature responsible for venetoclax resistance in different hematological malignancies [137]. In this respect, an in vitro analysis has demonstrated that the bromodomain and extra terminal proteins (BET) inhibitor JQ1 has an antitumor effect in CLL and, more importantly, that the combination of JQ1 and venetoclax enhances the apoptotic effect of the BCL2i [138]. These data point to the potential efficacy of BET inhibitors as a second-line treatment in case of venetoclax resistance. Additionally, because upregulation of MCL1 has been demonstrated to be involved in venetoclax resistance, the direct MCL1 antagonist AMG-176 has been shown to selectively kill CLL cells and act synergistically with venetoclax in vitro [139]. However, a phase I trial (NCT03797261) with the drug had to be discontinued due to safety concerns, leaving the door open for future MCL1 inhibitors development.

## 5. Biomarkers of Refractoriness to Immunotherapy

Cancer immunotherapy is a type of treatment that modulates the immune system in order to obtain an immune-mediated response against the tumor [140]. Immune checkpoint inhibitors and cellular immunotherapy for the treatment of CLL are currently under investigation, with conflicting results [141]. Due to its altered immune microenvironment and the promotion of immune dysfunction, CLL seemed to be an optimal candidate for immunotherapy, but for the same reasons, several difficulties have been encountered with immunotherapeutic approaches [141,142]. Several novel agents are under investigation for the immunotherapy of CLL, namely immune checkpoint inhibitors, chimeric antigen receptor (CAR)-T cells and bispecific/trispecific T/NK cells engagers [141]. Potential biomarkers of treatment refractoriness have been highlighted in recent years, including abnormal levels of circulating IL-10 and IL-6 and the reduced presence of specific memory T cell populations [24,47,48].

### 5.1. Abnormal IL-10 Levels

Higher circulating levels of IL-10 have been reported in advanced-stage CLL compared to early-stage and healthy patients [143]. IL-10 is an anti-inflammatory cytokine secreted by CLL neoplastic cells, and it is responsible, at least in part, for the suppression of the antitumor immune response observed in CLL through the intracellular signaling of its receptor IL-10R, expressed by various cell types, including T cells [144,145,146]. Recently, IL-10 has been proposed to be involved in refractoriness to immunotherapy with PD-1/PD-L1 immune checkpoint inhibitors [47].

The PD-1/PD-L1 axis is one of the most codified immune checkpoints, with an important role in the prevention of abnormal T-cell response [147]. PD-1 is a molecule expressed on the cellular membrane of normal T cells, which binds to its ligand PD-L1, physiologically found on the surface of antigen-presenting cells [148]. This interaction triggers the transduction of signaling pathways by the intracellular domain of PD-1, leading to the inhibition of PI3K/Akt and MAPK activation and finally resulting in T cell exhaustion and impaired function [147]. Since PD-1 and/or PD-L1 overexpression occurs in various tumors, the block of this axis by anti-PD1/PD-L1 mAbs has proven to be effective in several cancer types, including hematologic malignancies [149]. CLL patients overexpress PD-1 on the surface of T cells and PD-L1 on the external membrane of neoplastic B cells, thus providing an immunotolerant milieu that allows tumor escape from apoptosis [150].

Despite the apparently solid rationale for the treatment with immune checkpoint inhibitors, the results of a phase II study have shown the absence of efficacy of the anti-PD-1 mAb pembrolizumab, with an overall response rate (ORR) of 0% in relapsed CLL patients [151]. Because IL-10 serum levels are involved in refractoriness to immune checkpoint inhibitors, a combined approach of immune checkpoint inhibitors together with IL-10 inhibition has been attempted. Recently, initial promising results have been achieved with simultaneous treatment with IL-10-suppressing agents and pembrolizumab in CLL murine xenograft models [47]. The efficacy of pembrolizumab in reducing the tumor burden was amplified by 4.5-fold thanks to IL-10 suppression, paving the way for further investigations on the potential role of IL-10 as a predictive biomarker of resistance as well as a druggable target [47].

### 5.2. Potential Biomarkers of Refractoriness to CAR-T Cells

CAR-T cells are T lymphocytes extracted from the patient and engineered to express synthetic receptors, capable of specifically targeting and killing tumor cells through cytotoxic mechanisms [152]. After T cell extraction and ex vivo genetic engineering, CAR-T cells undergo the expansion process and are infused in the patient, who has been previously treated with a lymphodepleting conditioning regimen [153]. CARs consist of the fusion of an antigen-binding extracellular domain, a signaling CD3ζ subunit and one or more co-stimulatory intracellular domains [152,154]. Surprisingly, anti-CD19 CAR-T cell treatment has not reached the expected outcomes in CLL, with an average complete remission rate of ~30%, significantly lower compared to that obtained in acute lymphoblastic leukemia and diffuse large B cell lymphoma [155,156,157,158]. Several mechanisms of resistance to the treatment with anti-CD19 CAR-T cells in CLL have been proposed, principally due to T cell exhaustion promoted by the neoplastic clone [155]. The proposed mechanisms of resistance include impaired immune synapse formation, reduced expression of CD19 on the surface of CLL cells, production of tumor-derived extracellular vesicles, and higher expression of transmembrane inhibitory receptors on T cells, such as CTLA-4, LAG3, and PD-1 [159,160,161,162,163]. 

Recently, a genomic, phenotypic, and functional evaluation of CLL patients treated with anti-CD19 CAR-T cells has identified potential predictive biomarkers of response to treatment [48]. Higher blood levels of IL-6 correlated positively with response to treatment, making low serum IL-6 a biomarker of refractoriness to CAR-T cell-based immunotherapy [48]. Another candidate predictive biomarker was the presence of a significant population of CD27^+^CD45RO^−^ CD8^+^ T cells in responders before CAR-T cell generation, but further investigation is needed to determine the clinical value of these findings [48]. Since refractoriness to immunotherapy with CAR-T cells is primarily due to T cell exhaustion, immunotherapeutic approaches with CAR-NK cells are under evaluation for the treatment of CLL [155,164]. CAR-NK cells are engineered NK cells of healthy donors, extracted from umbilical cord blood or peripheral blood, designed to express on their surface a CAR receptor in order to take advantage of NK cytotoxic response against target tumor cells [165]. Although early promising results have been obtained in ongoing phase I/II trials (NCT03056339, NCT04796675, NCT04245722) with CAR-NK cells in CLL treatment, additional testing is required to assess the safety and efficacy profiles of this novel therapeutic approach [155].

## 6. Conclusions and Perspectives

During the last few years, several pathway inhibitors have been introduced for the treatment of CLL, leading, on the one hand, to chemo-free therapeutic strategies and higher rates of clinical response, and on the other hand, to the emergence of new mechanisms of refractoriness. Remarkably, treatment resistance to pathway inhibitors, namely BTKi and BCL2i, has been progressively clarified in patients treated with these innovative agents. The need to overcome these novel challenges has paved the way to understanding the detailed molecular mechanisms of refractoriness and developing new pathway inhibitors. The search for predictive biomarkers that might facilitate the identification of preclinical resistance in CLL treated with pathway inhibitors has also represented a matter of investigation, although the current guidelines do not recommend sequential testing of resistance mutations in patients who are responding to pathway inhibitors.

The imposition of continuous selection pressure through the administration of uninterrupted monotherapies with pathway inhibitors, in particular BTKi, is likely to facilitate the emergence of resistance mutations in a substantial proportion of patients. Consequently, the use of time-limited treatment approaches to avoid continued drug exposure and the selection of BTKi- and venetoclax-resistant clones represents a potential strategy to circumvent resistance development [166]. An encouraging strategy to overcome resistance once it has emerged is the development of next-generation BTKi that do not covalently bind to the target and are, therefore, still active in CLL cells, with the most common resistance mutations at the binding domain. In that regard, pirtobrutinib has shown to be effective in R/R CLL patients carrying BTKi mutations associated with refractoriness to covalent BTKi. At the same time, novel acquired mutations have been described to cause resistance to non-covalent BTKi, such as pirtobrutinib.

Finally, knowledge of biomarkers of refractoriness to CLL immunotherapy is still scant. Predictive biomarkers of response to treatment with CAR-T cells include levels of serum IL-6 and CD27^+^CD45RO^−^ CD8^+^ T cells, while no sufficient data are available on refractoriness to bispecific and trispecific T/NK cells engagers. Therefore, further investigations are needed to corroborate the initial findings of preclinical studies and early stage clinical trials.

## Figures and Tables

**Figure 1 ijms-24-10374-f001:**
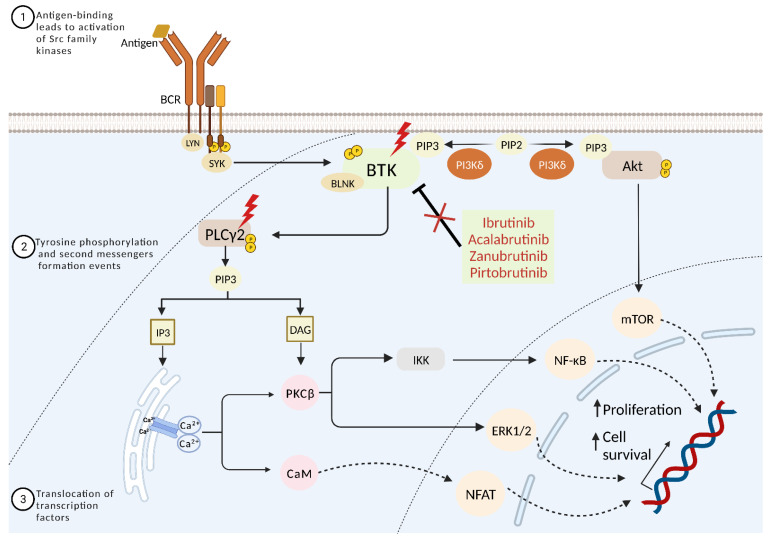
BCR signaling pathway and BTKis resistance in CLL. Upon antigen binding, the B cell receptor initiates the formation of a signaling complex through the phosphorylation of immunoreceptor-based activation motif (ITAM) residues on the cytoplasmic tails of CD79A(Igα) and CD79B(Igβ) proteins. This event activates SYK, which then triggers the activation of BTK, PLCγ2, and PI3K. The downstream signaling response includes PKC activation and Ca^2+^ mobilization and Akt activation, leading to the promotion of transcript factors NF-κB, ERK1/2, NFAT, and mTOR. This signaling cascade can be effectively inhibited by BTK inhibitors. However, BTK and PLCγ2 point mutations can result in BTKi resistance.

**Figure 2 ijms-24-10374-f002:**
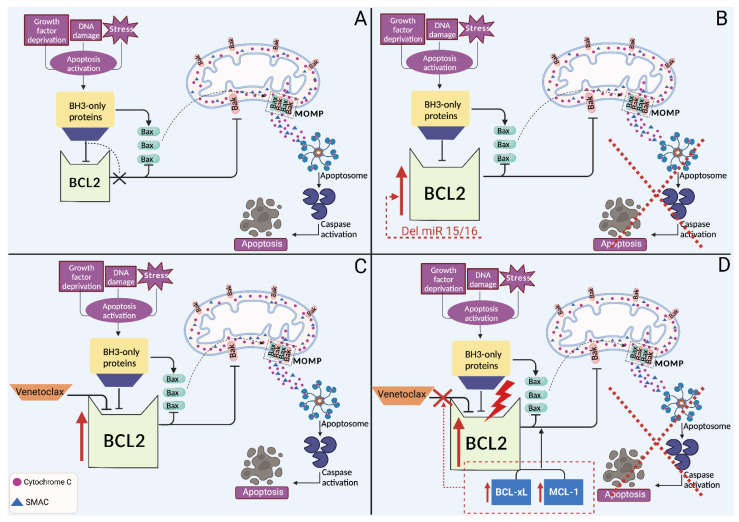
Intrinsic apoptosis pathway and venetoclax resistance in CLL. (**A**) Upon activation of the intrinsic apoptosis pathway by cellular stress, BH3-only proteins inhibit the anti-apoptotic proteins BCL2, BCL-xL, and MCL1. This inhibition leads to the activation and oligomerization of Bak and Bax, resulting in mitochondrial outer membrane permeabilization (MOMP). MOMP-mediated release of cytochrome c and SMAC leads to the formation of apoptosome, which results in the activation of caspase proteins leading to cell death. The anti-apoptotic BCL2 proteins inhibit this process by sequestering the pro-apoptotic proteins by binding to their BH3 motifs. (**B**) In CLL, BCL2 is overexpressed in 20–40% of the cases. The loss of miR-15 and miR-16 miRNAs leads to increased levels of BCL2, providing a survival advantage for the tumor. (**C**) Venetoclax induces apoptosis by binding to BCL2 protein, which is commonly found to be overexpressed in CLL. (**D**) Among venetoclax resistance mechanisms (i) BCL2 point mutations, which reduce the affinity for venetoclax (ii) upregulation of MCL-1 and BCL-xL abrogates venetoclax antitumor effect.

**Table 1 ijms-24-10374-t001:** Summary of identified biomarkers of treatment refractoriness to CLL.

Biomarker	Prevalence before Treatment	Prevalence at Progression	Mechanism of Resistance	Predictive Value	References
Unmutated *IGHV* gene	~40%	70–80%	Increased BCR signaling capacity	Poor response to CIT	[24,25,26]
Del(17p)	5%	~30%	Genomic instability, survival advantage, and reduced DNA damage response	Poor response to CIT, BTKi, and BCL2i	[24,25,27,28,29,30]
*TP53* mutations	7%	30–40%	Genomic instability, survival advantage, and reduced DNA damage response	Poor response to CIT, BTKi, and BCL2i	[24,25,27,28,29,30]
*BIRC3* mutations	2–6%	~8%	Upregulation of non-canonical NF-κB signaling pathway	Poor response to CIT	[24,31,32,33,34,35,36]
*NOTCH1* mutations	8–10%	30%	Transcriptional activation of cell survival and proliferation and reduced expression of CD20	Poor response to CIT and anti-CD20 mAbs	[24,25,27,37,38,39]
*BTK* point mutations of C481: C481S/R/Y/G	N/A	~50%	Reduced affinity for covalent BTKi	Poor response to covalent BTKi	[40]
*BTK* point mutations of the tyrosine kinase domain: L528W, V416L, T474I, M437R, A428D	N/A	~16%	Binding impairment of non-covalent BTKi	Poor response to covalent and non-covalent BTKi	[41]
*PLCG2* mutations: R665W, L845G, C849R, D993H	N/A	13%	Constitutively active PLCγ2	Poor response to BTKi	[42,43]
*BCL2* mutations: G101V, D103Y, F104I	N/A	~15%	Binding impairment of BCL2is	Poor response to BCL2i	[44]
Upregulation of MCL-1 and/or BCL-xL	N/A	N/A	Enhanced apoptosis evasion	Poor response to BCL2i	[45,46]
High serum [IL-10]	N/A	N/A	Reduced T cell response through IL-10R stimulation	Poor response to PD-1/PD-L1 immune checkpoint inhibitors	[47]
Low serum [IL-6]	N/A	N/A	CAR-T cell exhaustion due to defective IL-6R stimulation	Poor response to CAR-T cells	[48]
Low levels of CD27^+^CD45RO^−^ CD8^+^ T cells	N/A	N/A	Reduced population of active CAR-T cells	Poor response to CAR-T cells	[48]

Abbreviations: *IGHV*, immunoglobulin heavy variable; del(17p), deletion of the short arm of chromosome 17; BCR, B cell receptor; CIT, chemoimmunotherapy; *BTK*, Bruton tyrosine kinase; BTKi, BTK inhibitors; *BCL2*, B cell lymphoma 2; BCL2i, BCL2 inhibitors; PLCG2, phospholipase-C-gamma-2; PLCγ2, phospholipase-C-γ-2; IL-10, Interleukin-10; IL-10R, IL-10 receptor; IL-6, Interleukin-6; IL-6R, IL-6 receptor; CAR, chimeric antigen receptor; mAbs, monoclonal antibodies.

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
