# Peer review of "Treatment Refractoriness in Chronic Lymphocytic Leukemia: Old and New Molecular Biomarkers"

_ijms, 2023, doi:10.3390/ijms241210374_

Round 1

Reviewer 1 Report

Remarks about the manuscript: 

The authors have presented an excellent review article, "Treatment refractoriness in chronic lymphocytic leukemia: old and new molecular biomarkers." This article thoroughly examines the current treatment strategies and significant issues related to CLL.

- This review focuses on CLL and its treatment options.

- The review discusses the impact of immunotherapy on CLL.

- It also highlights the importance of tumor cells adapting to treatment. The assortment of references and updates in this manuscript has been thoughtfully selected.

The manuscript's English language is clear and flows smoothly. Based on the data in the manuscript, it is suitable for publication in the IJMS journal without any further changes. Therefore, I recommend it be published in its current form.

In my view, this manuscript (ID: IJMS-2425545) can be accepted without revision.

Author Response

Reviewer 1: The authors have presented an excellent review article, "Treatment refractoriness in chronic lymphocytic leukemia: old and new molecular biomarkers." This article thoroughly examines the current treatment strategies and significant issues related to CLL.

- This review focuses on CLL and its treatment options.

- The review discusses the impact of immunotherapy on CLL.

- It also highlights the importance of tumor cells adapting to treatment. The assortment of references and updates in this manuscript has been thoughtfully selected.

The manuscript's English language is clear and flows smoothly. Based on the data in the manuscript, it is suitable for publication in the IJMS journal without any further changes. Therefore, I recommend it be published in its current form.

In my view, this manuscript (ID: IJMS-2425545) can be accepted without revision.

 We give thanks to the Reviewer for his/her appreciative words in regard of our work.

Reviewer 2 Report

The manuscript entitled "Treatment refractoriness in chronic lymphocytic leukemia: old and new molecular biomarkers" is a very interesting and very well-written paper.

However, the introduction should contain more information about leukemia, for example, information about its history.

In addition, a new subsection should be included at the end of the manuscript, but before the conclusion, in which the authors would have described past and present treatments for leukemia. Chemotherapy is mentioned in the manuscript but rather in a minor way, because in the early stages leukaemia was mainly treated with chemotherapy, for instance using busulfan. The information of the latter should be given and described.

The manuscript should refer to data from the International Agency for Research on Cancer, including information on the rising incidence and mortality rates of the disease.

Author Response

Reviewer 2: The manuscript entitled "Treatment refractoriness in chronic lymphocytic leukemia: old and new molecular biomarkers" is a very interesting and very well-written paper.

 However, the introduction should contain more information about leukemia, for example, information about its history.

 In addition, a new subsection should be included at the end of the manuscript, but before the conclusion, in which the authors would have described past and present treatments for leukemia. Chemotherapy is mentioned in the manuscript but rather in a minor way, because in the early stages leukaemia was mainly treated with chemotherapy, for instance using busulfan. The information of the latter should be given and described.

 The manuscript should refer to data from the International Agency for Research on Cancer, including information on the rising incidence and mortality rates of the disease.

  1. We thank the Reviewer for his/her constructive comment. Additional information about chronic lymphocytic leukemia (CLL) and its history has been added to the manuscript (lines 37-41).
  2. We have added an overview of past chemotherapy treatment for CLL, in order to address the comment made by the Reviewer (lines 103-107). Since busulfan was used in chronic myeloid leukemia and is currently adopted in conditioning regimen for the treatment of acute leukemia, we did not include this treatment in our work, which refers only to CLL.
  3. Unfortunately, no data on CLL epidemiology are available from the International Agency for Research on Cancer (IARC). The only available data refer to leukemia in general. However, we have used the data from the Surveillance, Epidemiology and End Results Program (SEER), which provides information about incidence and mortality of CLL in the United States (line 45).

Round 2

Reviewer 2 Report

I recommend the article for publication in its current form.